# Analytic Formulae for T Violation in Neutrino Oscillations

**DOI:** 10.3390/e26060472

**Published:** 2024-05-29

**Authors:** Osamu Yasuda

**Affiliations:** Department of Physics, Tokyo Metropolitan University, Tokyo 192-0397, Japan; yasuda@phys.se.tmu.ac.jp

**Keywords:** neutrino oscillation, T violation, *μ*TRISTAN

## Abstract

Recently, a concept known as μTRISTAN, which involves the acceleration of μ+, has been proposed. This initiative has led to considerations of a new design for a neutrino factory. Additionally, leveraging the polarization of μ+, measurements of T violation in neutrino oscillations are also being explored. In this paper, we present analytical expressions for T violation in neutrino oscillations within the framework of standard three-flavor neutrino oscillations, a scenario involving nonstandard interactions, and a case of unitarity violation. We point out that examining the energy spectrum of T violation may be useful for probing new physics effects.

## 1. Introduction

Results from various neutrino oscillation experiments have nearly determined the three mixing angles and the absolute values of the mass squared differences in the standard three-flavor mixing scenario within the lepton sector [1]. The remaining undetermined parameters, such as the mass ordering, the octant of the atmospheric neutrino oscillation mixing angle, and the CP phase, are expected to be resolved by the high-intensity neutrino long-baseline experiments currently under construction, such as T2HK and DUNE. Once the CP phase is established, the standard three-flavor lepton mixing scheme will be solidified, achieving the final goal of studies on standard three-flavor neutrino oscillations. To explore physics beyond this framework using neutrino oscillations, experiments in previously unexplored channels will be necessary.

Recently, a concept known as μTRISTAN [2] has been proposed, which involves creating a low-emittance μ+ beam using ultra-cold muon technology and accelerating it to energies suitable for a μ+ collider. The expected number of muons at μTRISTAN is on the order of 1013 to 1014 muons per second. This proposal has reignited interest [3] in the neutrino factory concept [4,5], which could be developed en route to achieving a muon collider. At such a neutrino factory, the decay of μ+ in the storage ring would produce ν¯μ and νe. Ref. [6] explored the potential to polarize the μ beam to reduce the flux of νμ or ν¯μ, thereby enabling the measurement of νe→νμ transitions. If this can be achieved, it would allow for the measurement of T violation in neutrino oscillations, i.e., the difference between the oscillation probabilities P(νμ→νe) and P(νe→νμ).

T violation in neutrino oscillations has been discussed by many researchers in the past [6,7,8,9,10,11,12,13,14,15,16,17,18,19,20,21,22,23,24,25,26,27,28]. T violation has attracted significant attention primarily because its structure is simpler than that of CP violation, which compares P(νμ→νe) with P(ν¯μ→ν¯e) and involves complications due to the presence of the matter effect. (To justify discussions of T violation on an equal footing with CP violation, CPT symmetry is necessary in the neutrino sector. Refs. [29,30,31,32,33] studied CPT symmetry in the neutrino sector and concluded that there are strong constraints on CPT violation.) In this paper, we derive the analytical forms of T violation in three scenarios: the standard three-flavor scheme, a scenario with nonstandard interactions, and a case of unitarity violation. We also briefly comment on the feasibility of probing new physics effects by examining the energy dependence of T violation.

In Section 2, we review the formalism by Kimura, Takamura, and Yokomakura [34,35] to derive analytical formulas for the oscillation probabilities. In Section 3, we derive the analytic forms for T violation in the three cases: the standard three-flavor mixing framework, a scenario involving flavor-dependent nonstandard interactions, and a case with unitarity violation. In Section 4, we summarize our conclusions.

## 2. Analytical Formula for Oscillation Probabilities

It is known [36] (see also earlier works [37,38,39]) that after eliminating the negative energy states by a Tani–Foldy–Wouthusen-type transformation, the Dirac equation for neutrinos propagating in matter is reduced to the familiar form:(1)idΨdt=UEU−1+AΨ,
where *U* is the PMNS matrix,
Ψ≡νeνμντ
is the flavor eigenstate,
(2)E≡diagE1,E2,E3
is the diagonal matrix of the energy eigenvalue Ej≡mj2+p→2(j=1,2,3) of each mass eigenstate with momentum p→, and the matrix
A≡2GFdiagNe−Nn2,−Nn2,−Nn2.
stands for the matter effect, which is characterized by the Fermi coupling constant GF, the electron density Ne, and the neutron density Nn. Throughout this paper we assume for simplicity that the density of matter is constant. The 3×3 matrix on the right-hand side of Equation (Equation 1) is Hermitian and can be formally diagonalized by a unitary matrix U˜ as
(3)UEU−1+A=U˜E˜U˜−1,
where
E˜≡diagE˜1,E˜2,E˜3
is a diagonal matrix with the energy eigenvalue E˜j in the presence of the matter effect. Equation (Equation 1) can be easily solved, resulting in the flavor eigenstate at the distance *L*:(4)Ψ(L)=U˜exp−iE˜LU˜−1Ψ(0).
Thus, we have the probability amplitude A(νβ→να) of the flavor transition νβ→να:(5)A(νβ→να)=U˜exp−iE˜LU˜−1αβ.
From Equation (Equation 5) we observe that the shift E→E−1E1, where 1 stands for the 3×3 identity matrix, changes only the overall phase of the probability amplitude A(νβ→να), and this phase does not affect the value of the probability P(νβ→να)=|A(νβ→να)|2 of the flavor transition νβ→να. In the following discussions, therefore, for simplicity, we define the diagonal energy matrix E and the potential one A as follows:
(6)E≡diagE1,E2,E3−E11=diag0,ΔE21,ΔE31A≡2GFdiagNe−Nn2,−Nn2,−Nn2+Nn21
(7)=diagA,0,0
with
ΔEjk≡Ej−Ek≃mj2−mk22|p→|≡Δmjk22|p→|≡Δmjk22E
(8)A≡2GFNe.
Thus, the appearance of the oscillation probability P(νβ→να)(α≠β) is given by
P(νβ→να)=U˜exp−iE˜LU˜−1αβ2=∑j=13X˜jαβe−iE˜jL2=e−iE˜1L∑j=13X˜jαβe−iΔE˜j1L2
(9)=∑j=13X˜jαβe−iΔE˜j1L−12
=(−2i)∑j=23e−iΔE˜j1L/2X˜jαβsinΔE˜j1L22=4e−iΔE˜31L/2X˜3αβsinΔE˜31L2+e−iΔE˜21L/2X˜2αβsinΔE˜21L22
(10)=4X˜3αβsinΔE˜31L2+eiΔE˜32L/2X˜2αβsinΔE˜21L22
where
X˜jαβ≡U˜αjU˜βj*ΔE˜jk≡E˜j−E˜k
have been defined,
(11)∑j=13X˜jαβ=δαβ=0forα≠β
was subtracted in Equation (Equation 9) and throughout this paper the indices α,β=(e,μ,τ) and j,k=(1,2,3) stand for those of the flavor and mass eigenstates, respectively. Once we know the eigenvalues E˜j and the quantity X˜jαβ, the oscillation probability can be expressed analytically. ( In the case of three neutrino flavors in matter, the energy eigenvalues, E˜j, can in principle be analytically determined using the cubic equation root formula [40]. However, the analytic expression for E˜j involving the inverse cosine function is not practically useful. Therefore, below we will calculate E˜j using perturbation theory with small parameters, such as ΔE21/ΔE31=Δm212/Δm312≈1/30 and those relevant to nonstandard scenarios). So the only non-trivial problem in the standard case is to obtain the expression for X˜jαβ, and this was achieved by Kimura, Takamura and Yokomakura [34,35]. Their arguments are based on the trivial identities. From the unitarity condition of the matrix U˜, we have
(12)δαβ=U˜U˜−1αβ=∑jU˜αjU˜βj*=∑jX˜jαβ.
Next, we take the (α,β) component of both sides in Equation (Equation 3):
(13)UEU−1+Aαβ=U˜E˜U˜−1αβ=∑jU˜αjE˜jU˜βj*=∑jE˜jX˜jαβ
Furthermore, we take the (α,β) component of the square of Equation (Equation 3):
(14)UEU−1+A2αβ=U˜E˜2U˜−1αβ=∑jU˜αjE˜j2U˜βj*=∑jE˜j2X˜jαβ
Putting Equations (12)–(14) together, we have
(15)111E˜1E˜2E˜3E˜12E˜22E˜32X˜1αβX˜2αβX˜3αβ=Y1αβY2αβY3αβ
with
Yjαβ≡UEU−1+Aj−1αβforj=1,2,3,
which can be easily solved by inverting the Vandermonde matrix:
(16)X˜1αβX˜2αβX˜3αβ=1ΔE˜21ΔE˜31(E˜2E˜3,−(E˜2+E˜3),1)−1ΔE˜21ΔE˜32(E˜3E˜1,−(E˜3+E˜1),1)1ΔE˜31ΔE˜32(E˜1E˜2,−(E˜1+E˜2),1)Y1αβY2αβY3αβ.

## 3. Analytic Form of T Violation

In this section, we derive the analytic form of T violation in the cases with and without unitarity, using the formalism described in Section 2.

### 3.1. The Three-Flavor Case with Unitarity

First, let us discuss the case where time evolution is unitary. From Equation (10), we have
(17)P(νμ→νe)−P(νe→νμ)=4sinΔE˜31L2sinΔE˜21L2×eiΔE˜32L/2X˜3eμ*X˜2eμ+e−iΔE˜32L/2X˜3eμX˜2eμ*−eiΔE˜32L/2X˜3eμX˜2eμ*−e−iΔE˜32L/2X˜3eμ*X˜2eμ=16ImX˜2eμX˜3eμ*sinΔE˜32L2sinΔE˜31L2sinΔE˜21L2,
Furthermore, from Equation (Equation 16), the factor ImX˜2eμX˜3eμ* in Equation (17) can be rewritten as
(18)ImX˜2eμX˜3eμ*=−1ΔE˜21ΔE˜321ΔE˜31ΔE˜32×Im{Y3eμ−(E˜3+E˜1)Y2eμ}{Y3eμ*−(E˜1+E˜2)Y2eμ*}=1ΔE˜21ΔE˜31ΔE˜32ImY2eμY3eμ*
Equations (17) and (18) are applicable for a generic case, as long as the unitarity relation (11) holds.

#### 3.1.1. The Standard Three-Flavor Case

In the standard three-flavor case, Yj+1αβ≡[(UEU−1+A)j]αβ (j=1,2) can be expressed as follows:
Y2αβ≡UEU−1+Aαβ
(19)=∑j=23ΔEj1Xjαβ+Aδαeδβe
Y3αβ≡UEU−1+A2αβ
(20)=∑j=23(ΔEj1)2Xjαβ+A∑j=23ΔEj1δαeXjeβ+δβeXjαe+A2δαeδβe,
where we have also defined the quantity in vacuum:(21)Xjαβ≡UαjUβj*.
From Equations (19) and (20), the factor ImY2eμY3eμ* in Equation (18) can be rewritten as
(22)ImY2eμY3eμ*=ImΔE21X2eμ+ΔE31X3eμ×ΔE21ΔE21+AX2eμ*+ΔE31ΔE31+AX3eμ*=ImX2eμX3eμ*ΔE21ΔE31ΔE32
ImX2eμX3eμ* in Equation (22) is the Jarlskog factor [41] for the lepton sector, and is given in the standard parametrization [1] with the three mixing angles θjk(j,k=1,2,3) and the Dirac CP phase δ by
J≡ImX2eμX3eμ*=−18sinδcosθ13sin2θ12sin2θ13sin2θ23.
Hence, we obtain
(23)P(νμ→νe)−P(νe→νμ)=16JΔE21ΔE31ΔE32ΔE˜21ΔE˜31ΔE˜32sinΔE˜32L2sinΔE˜31L2sinΔE˜21L2
Equation (23) is a well-known formula [10] for the standard three-flavor case. It is remarkable that in the standard three-flavor case, T violation in matter, when divided by the δ-independent factor 16∏j>k[sin(ΔE˜jkL/2)ΔEjk/ΔE˜jk], coincides with the Jarlskog factor in vacuum. This implies that the only source of T violation is the CP phase δ in the standard three-flavor case, and it is the reason why T violation is simpler than CP violation in neutrino oscillations.

#### 3.1.2. The Case with Nonstandard Interactions

As long as unitarity in the three-flavor framework is maintained, Equation (18) holds. In this subsection, let us consider the scenario with flavor-dependent nonstandard interactions [42,43] during neutrino propagation. This scenario has garnered significant attention due to its potential implications for phenomenology. In this case, the mass matrix is given by
(24)UEU−1+A+ANP
with
ANP≡Aϵeeϵeμϵeτϵeμ*ϵμμϵμτϵeτ*ϵμτ*ϵττ,
where A and *A* are given by Equations (7) and (8), respectively. The dimensionless quantities ϵαβ stand for the ratio of the nonstandard Fermi coupling constant interaction to the standard one. Since the matrix (24) is Hermitian, time evolution is unitary and all the arguments up to Equation (18) hold also in this case. The oscillation probability is given by Equations (10) and (16), where the standard potential matrix A must be replaced by A+ANP.

The extra complication compared to the standard case is calculations of the eigenvalues E˜j and the elements [(UEU−1+A+ANP)m]αβ (m=1,2). Here, we work with perturbation theory with respect to the small parameters ΔE21/ΔE31=Δm212/Δm312≃1/30 and ϵαβ, which we assume to be as small as ΔE21/ΔE31. Namely, throughout this paper we assume
|ΔE31|∼A≫|ΔE21|≳A|ϵαβ|.
and take into consideration to first order in these small parameters. Then, to first order in them, we obtain
Y2eμ≡UEU−1+A+ANPeμ
(25)=ΔE31X3eμ+ΔE21X2eμ+Aϵeμ
Y3eμ≡UEU−1+A+ANP2eμ≃{(ΔE31)2+AΔE31}X3eμ+AΔE21X2eμ+A2ϵeμ+ΔE31X3,ANPeμ
(26)=(ΔE31+A)Y2eμ−ΔE31ΔE21X2eμ−AΔE31ϵeμ+ΔE31X3,ANPeμ,
where the curly bracket stands for an anticommutator of matrices *P* and *Q*: {P,Q}≡PQ+QP, and X3 is a 3×3 matrix defined by
(27)X3≡Udiag0,0,1U−1X3αβ=Uα3Uβ3*=X3αβ.
From this, the first term on the right-hand side of Equation (26) drops in the factor ImY2eμY3eμ*, and we obtain
ImY2eμY3eμ*=−ImY2eμ*Y3eμ≃ΔE31ImY2eμ*ΔE21X2eμ+Aϵeμ−X3,ANPeμ≃(ΔE31)2ImX3eμ*ΔE21X2eμ+Aϵeμ−X3,ANPeμ=(ΔE31)2ImX3eμ*ΔE21X2eμ+AX3ττϵeμ−X3eτϵτμ−X3τμϵeτ,
where we have ignored terms of order O((ΔE21)2), O(A2(ϵαβ)2), and O(AΔE21ϵαβ). Thus, we finally obtain the form for T violation:(28)P(νμ→νe)−P(νe→νμ)≃16(ΔE31)2ΔE˜21ΔE˜31ΔE˜32sinΔE˜32L2sinΔE˜31L2sinΔE˜21L2×ImX3eμ*ΔE21X2eμ+AX3ττϵeμ−X3eτϵτμ−X3τμϵeτ
Note that the form of the standard contribution (ΔE31)2ΔE21,ImX2eμX3eμ* in Equation (28) differs slightly from that in Equation (22) because we are neglecting terms of order O((ΔE21)2). The terms proportional to *A* in the parenthesis in Equation (28) represent the additional contributions to T violation due to nonstandard interactions. These additional contributions are constant with respect to the neutrino energy *E*, and they exhibit a different energy dependence from that of the standard one, ΔE21X2eμ=(Δm212/2E)Ue2Uμ2*. Therefore, if the magnitude of the additional contributions from nonstandard interactions is significant enough, then their effects are expected to be observable in the energy spectrum of T violation. Constraints on the parameters ϵαβ have been provided in Refs. [44,45,46]. Depending on the sensitivity of each experiment, it may or may not be possible to detect the signal or to improve the existing bounds on ϵαβ. The aim of this paper is to derive the analytic form of T violation; estimating experimental sensitivity is beyond its scope.

### 3.2. The Three-Flavor Case with Unitarity Violation

The discussions in Section 3.1 are based on the assumption that time evolution is unitary. In Ref. [47], the possibility to have a nonunitary leptonic mixing matrix was pointed out. In that case, the relation between the mass eigenstate νj and the flavor eigenstate να is given by a nonunitary matrix *N*:
να=NαjνjN≡(1+η)U
with
(29)η†=η.
In the so-called minimal unitarity violation, which was discussed in Ref. [47], the constraint on the deviation matrix η turned out to be strong. Here, we take phenomenologically the form of the nonunitary matrix *N* and assume that the elements of the deviation matrix η are of order ΔE21/ΔE31 or smaller, as in Section 3.1.2, namely,
|ΔE31|∼A≫|ΔE21|≳A|ηαβ|.
It was argued in Ref. [48] that time evolution in the case of a nonunitary mixing matrix can be discussed in terms of the mass eigenstate
Ψm≡ν1ν2ν3,
and its time evolution is described by
(30)idΨmdt=E+NTAN*−AnNTN*−1Ψm,
where E and A are defined by Equations (Equation 6) and (7),
An≡12GFNn
stands for the absolute value of the contribution to the matter effect from the neutral current interaction, and the term An1 was added to simplify the calculations without changing the absolute value of the probability amplitude. The 3×3 matrix on the right-hand side of Equation (30) can be diagonalized with a unitary matrix *W*:(31)E+NTAN*−AnNTN*−1=WE˜W−1,
where
E˜≡diagE˜1,E˜2,E˜3
is the energy eigenvalue matrix in matter with unitarity violation. The mass eigenstate at distance *L* can be solved as
(32)Ψm(L)=Wexp(−iE˜L)W−1Ψm(0).
In cases involving unitarity violation, due to the modified form of the charged current interaction [47], after computing the probability amplitude from Equation (32) we must multiply the probability amplitude by an additional factor of (NN†)ββ−1/2 for the production process and (NN†)αα−1/2 for detection. Defining the modified amplitude
A^(νβ→να)≡A(νβ→να)(NN†)αα1/2(NN†)ββ1/2=[N*Wexp(−iE˜L)W−1NT]αβ,
the modified probability
P^(να→νβ)≡|A^(να→νβ)|2,
and the quantity
X˜jαβ≡(N*W)αj(NW*)βj,
we have the following expression for the appearance oscillation probability:(33)P^(νβ→να)=N*Wexp−iE˜LW−1NTαβ2=∑j=13X˜jαβe−iE˜jL2=e−iE˜1L∑j=13X˜jαβe−iΔE˜j1L2=∑j=13X˜jαβ1−1−e−iΔE˜j1L2=[N*NT]αβ−2i∑j=23e−iΔE˜j1L/2X˜jαβsinΔE˜j1L22=[{(1+η)2}T]αβ+2e−iΔE˜31L/2−iπ/2X˜3αβsinΔE˜31L2+2e−iΔE˜21L/2−iπ/2X˜2αβsinΔE˜21L22≃4ηβα+e−iΔE˜31L/2−iπ/2X˜3αβsinΔE˜31L2+e−iΔE˜21L/2−iπ/2X˜2αβsinΔE˜21L22.
T violation P(νμ→νe)−P(νe→νμ) is a small quantity, and the difference between the probability P(νμ→νe) and the modified one P^(νμ→νe) comes from the factor (NN†)αα(NN†)ββ=[(1+η)2]αα[(1+η)2]ββ≃1+2ηαα+2ηββ, which has a small deviation from 1. Therefore, T violation of the probability P(νμ→νe)−P(νe→νμ) can be approximated by that of the modified probability P^(νμ→νe)−P^(νe→νμ). Hence, T violation is given by
(34)P(νμ→νe)−P(νe→νμ)≃P^(νμ→νe)−P^(νe→νμ)=4ημe+e−iΔE˜31L/2−iπ/2X˜3eμsinΔE˜31L2+e−iΔE˜21L/2−iπ/2X˜2eμsinΔE˜21L22−4ηeμ+e−iΔE˜31L/2−iπ/2X˜3μesinΔE˜31L2+e−iΔE˜21L/2−iπ/2X˜2μesinΔE˜21L22=4sinΔE˜31L2ημeX˜3eμ*−ημe*X˜3eμ(eiΔE˜31L/2+iπ/2−e−iΔE˜31L/2−iπ/2)+4sinΔE˜21L2ημeX˜2eμ*−ημe*X˜2eμ(eiΔE˜21L/2+iπ/2−e−iΔE˜21L/2−iπ/2)−4sinΔE˜31L2sinΔE˜21L2X˜3eμX˜2eμ*−X˜3eμ*X˜2eμ×(eiΔE˜31L/2+iπ/2−iΔE˜21L/2−iπ/2−e−iΔE˜31L/2−iπ/2+iΔE˜21L/2+iπ/2)=−16ImX˜2eμX˜3eμ*sinΔE˜31L2sinΔE˜21L2sinΔE˜32L2+8ImημeX˜3eμ*sinΔE˜31L+8ImημeX˜2eμ*sinΔE˜21L
We observe that the energy dependence of T violation in this case is different from that with unitarity, since we have extra contributions which are proportional to sinΔE˜31L or sinΔE˜21L. As in the case with unitarity, X˜jαβ can be expressed in terms of the quantity Xjαβ≡UαjUβj* in vacuum, E˜j, and ηαβ. First of all, we note the following relations:(35)∑j(E˜j)mX˜jαβ=∑j(N*W)αj(E˜j)m(NW*)βj=N*E+NTAN*−AnNTN*−1mNTαβ=NE+N†AN−AnN†N−1mN†βα≡Ym+1αβform=0,1,2.
Then, we rewrite Equation (Equation 35) as
(36)∑m=13VjmX˜mαβ=Yjαβforj=1,2,3,
where Vjm≡(E˜m)j−1 is the element of the Vandermonde matrix *V*, as in the case with unitarity (see Equation (15)). The simultaneous Equation (36) can be solved by inverting *V*, and we obtain
(37)X˜jαβ=∑m=13(V−1)jmYmαβ.
The factor ImX˜2eμX˜3eμ* can be expressed in terms of E˜j and Yjeμ:(38)ImX˜2eμX˜3eμ*=−1ΔE˜21ΔE˜321ΔE˜31ΔE˜32×Im{Y3eμ−(E˜3+E˜1)Y2eμ+E˜3E˜1Y1eμ}{Y3eμ*−(E˜1+E˜2)Y2eμ*+E˜1E˜2Y1eμ*}=1ΔE˜21ΔE˜31ΔE˜32E˜12ImY1eμY2eμ*−E˜1ImY1eμY3eμ*+ImY2eμY3eμ*.
The quantities Yjeμ(j=1,2,3) are calculated as follows:
Y1eμ=[NN†]μe=[(1+η)2]μe≃2ημe,
Y2eμ=NE+N†AN−AnN†N−1N†μe=(1+η)UEU−1+(1+η)A(1+η)−An(1+η)2−1(1+η)μe≃UEU−1+A+{η,UEU−1}+2{η,A}−2Anημe≃ΔE31X3μe+ΔE21X2μe+ΔE31{η,X3}μe+2(A−An)ημe,
Y3eμ=NE+N†AN−AnN†N−12N†μe=(1+η)UEU−1+(1+η)A(1+η)−An(1+η)2−12(1+η)μe≃UE2U−1+A2+{UEU−1,A}μe+{UEU−1,{η,A}}+{A,{η,A}}μe−2An{UEU−1,η}+2An{A,η}μe+{η,UE2U−1}+{η,A2}+{η,{UEU−1,A}}μe≃ΔE31(ΔE31+A)X3μe+AΔE21X2μe+ΔE31{X3,{η,A}}μe+{A,{η,A}}μe−2AnΔE31{X3,η}μe−2An{A,η}μe+ΔE312{η,X3}μe+{η,A2}μe+ΔE31{η,{X3,A}}μe=(ΔE31+A)Y2eμ−ΔE31ΔE21X2μe−2(A−An)ΔE31ημe−(A+2An)ΔE31{X3,η}μe+ΔE31{X3,{η,A}}μe+ΔE31{η,{X3,A}}μe.
In the current scenario involving unitarity violation, we observe a nonvanishing contribution from Y1αβ, necessitating knowledge of the explicit form of the energy eigenvalue E˜1. Given that Y1eμ is of order O(ηαβ), to evaluate Equation (38) accurately to first order in both ΔE21/ΔE31 and ηαβ, we must calculate E˜1 solely to zeroth order in these parameters, i.e., assuming ΔE21→0 and ηαβ→0. Under these conditions, the characteristic equation of the 3×3 matrix (31) is defined by
0=det1t−E+NTAN*−AnNTN*−1≃det1t−diag(0,0,ΔE31)−U−1diag(A,0,0)U=tt2−(ΔE31+A)t+AΔE31cos2θ13=t(t−λ+)(t−λ−),
where λ± are the roots of the quadratic equation and are given by
λ±≡ΔE31+A±ΔE˜312ΔE˜31≡(ΔE31cos2θ13−A)2+(ΔE31sin2θ13)2.
From this, we obtain the energy eigenvalues E˜j(j=1,2,3) to the leading order in ΔE21/ΔE31 and ηαβ:E˜1E˜2E˜3≃λ−0λ+
The roots λ−=E˜1 and λ+=E˜3 satisfy the quadratic equation
λ±2−(ΔE31+A)λ±+AΔE31cos2θ13=0.
Hence, the first two terms on the right-hand side of Equation (38) can be rewritten as
E˜12ImY1eμY2eμ*−E˜1ImY1eμY3eμ*=ImY1eμE˜12Y2eμ*−E˜1Y3eμ*≃ImY1eμΔE31X3μe*E˜12−ΔE31(ΔE31+A)X3μe*E˜1=ImY1eμA(ΔE31)2X3μe*cos2θ13=2A(ΔE31)2cos2θ13ImημeX3μe*,
whereas the third term on the right-hand side of Equation (38) can be written as
ImY2eμY3eμ*=−ImY2eμ*(ΔE31+A)Y2eμ−ΔE31ΔE21X2μe−2(A−An)ΔE31ημe−(A+2An)ΔE31{X3,η}μe+ΔE31{X3,{η,A}}μe+ΔE31{η,{X3,A}}μe=ΔE31ImX3μe*ΔE31ΔE21X2μe+2(A−An)ΔE31ημe+(A+2An)ΔE31{X3,η}μe−ΔE31{X3,{η,A}}μe−ΔE31{η,{X3,A}}μe.
Thus, we obtain the expression for the factor ImX˜2eμX˜3eμ*:(39)ImX˜2eμX˜3eμ*=1ΔE˜21ΔE˜31ΔE˜32E˜12ImY1eμY2eμ*−E˜1ImY1eμY3eμ*+ImY2eμY3eμ*≃ΔE31ΔE˜21ΔE˜31ΔE˜32ImX3μe*ΔE21X2μe+4Aημe+2An{η,X3}μe−ημe
To complete the calculation of Equation (34), we need to estimate the two quantities:(40)ImημeX˜3eμ*=1ΔE˜31ΔE˜32ImημeE˜1E˜2Y1eμ*−(E˜1+E˜2)Y2eμ*+Y3eμ*≃1ΔE˜31ΔE˜32Imημe−E˜1Y2eμ*+Y3eμ*≃1ΔE˜31λ+Imημe−λ−ΔE31X3μe*+ΔE31(ΔE31+A)X3μe*=1ΔE˜31λ+λ+ΔE31ImημeX3μe*=ΔE31ΔE˜31ImημeX3μe*,
(41)ImημeX˜2eμ*=−1ΔE˜21ΔE˜32ImημeE˜3E˜1Y1eμ*−(E˜3+E˜1)Y2eμ*+Y3eμ*≃−1ΔE˜21ΔE˜32Im−(E˜3+E˜1)Y2eμ*+Y3eμ*≃−1ΔE˜21ΔE˜32Im−(E˜3+E˜1)ΔE31X3μe*+ΔE31(ΔE31+A)X3μe*≃0,
where terms of order O((ΔE21/ΔE31)2), O((ϵαβ)2), and O(ϵαβΔE21/ΔE31) have been neglected in Equations (39)–(41). Putting Equations (39)–(41) together, the final expression for T violation is given by
(42)P(νμ→νe)−P(νe→νμ)≃16ΔE31AΔE˜31cos2θ13sinΔE˜31L2sinΔE˜21L2sinΔE˜32L2×ImX3μe*ΔE21X2μe+4Aημe+2An{η,X3}μe−ημe+8ΔE31ΔE˜31sinΔE˜31LImημeX3μe*
Due to the additional contribution proportional to sin(ΔE˜31L), the energy dependence of Equation (42) in the scenario with unitarity violation differs from that in the scenarios with unitarity, such as the standard case (23) and the nonstandard interaction case (42). Therefore, if the contribution from unitarity violation is significant enough and the experimental sensitivity is sufficiently high, it may be possible to distinguish the unitarity violation scenario from both the standard and nonstandard interaction scenarios by examining the energy spectrum in T violation.

## 4. Conclusions

In this paper, we have derived the analytical expression for T violation in neutrino oscillations under three different scenarios: the standard three-flavor mixing framework, a scenario involving flavor-dependent nonstandard interactions, and a case with unitarity violation. In scenarios preserving unitarity, the T-violating component of the oscillation probability is proportional to sin(ΔE˜31L2)sin(ΔE˜21L2)sin(ΔE˜32L2). However, in the case with unitarity violation, there is an additional contribution proportional to sin(ΔE˜31L). Should future long-baseline experiments, such as μTRISTAN or other types of neutrino factories, achieve high sensitivity to T violation across a broad energy spectrum, it may become feasible to specifically probe unitarity in the νμ↔νe channel.

Moreover, we demonstrated that the coefficient of the term sin(ΔE˜31L2)sin(ΔE˜21L2) sin(ΔE˜32L2)(ΔE31)2(ΔE˜31ΔE˜32ΔE˜21)−1 varies depending on whether neutrino propagation follows the standard scheme or involves nonstandard interactions. In the standard scenario, this coefficient is proportional to ΔE21=Δm212/2E. However, in the case with nonstandard interactions, there is an additional contribution that is energy-independent. Thus, it may be possible to observe the effects of nonstandard interactions by examining the energy dependence of T violation.

The purpose of this paper is to derive the analytical expression of T violation, and we did not quantitatively discuss the sensitivity of future experiments. The potential for T violation in neutrino oscillations deserves further study.

## Data Availability

Data is contained within the article.

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
