# Peer review of "Analytic Formulae for T Violation in Neutrino Oscillations"

_entropy, 2024, doi:10.3390/e26060472_

Round 1
Reviewer 1 Report
Comments and Suggestions for Authors
Dear Yasuda:
Your article regarding T-violation expressions in neutrino oscillations is well-written and sound. Here are some comments on the article:
- On the introduction you say: "concluding the studies of the Standard Model with three massive neutrinos." I understand that you mean to conclude the studies of the parameters associated to neutrino oscillations; however, that is not the end of the story. We need to understand the origin and nature of the neutrino mass, as well as its absolute mass scale. I encourage you to be more precise with the wording here.
- I found muTristran an interesting idea; however, not enough details are given to state its experimental reach. Please add some benchmark numbers of muTristran performance, e.g., expected rates or neutrino luminosity.
- After equation (23), you say that "it's remarkable that the T violation is proportional to the Jarlskog factor in a vacuum." This is true, though the amplitude of the T-violation is still matter dependent since you have the ratio of the Delta E to Delta E\tilde. So, the extraction of the Jarlskog invariant through this method would still be matter-effect dependent. Please make the phrasing more accurate.
- I think a brief discussion (one or two sentences) on current constraints of CPT symmetry with neutrinos should be present in the introduction, since this is needed to connect T and CP oscillation probabilities.
- I think the paper would benefit with some figures that can compare the results of Eqs. (23), (28), and (42) for benchmark values of epsilon and eta.
- on the conclusion, it would be nice if the author echoed the experimental programs that can measure this effect.
Author Response
Dear Referee,
Thank you for reviewing my work and for giving me valuable
comments. Below we give my reply to the comments:
> - On the introduction you say: "concluding the studies of the
> Standard Model with three massive neutrinos." I understand that
> you mean to conclude the studies of the parameters associated to
> neutrino oscillations; however, that is not the end of the
> story. We need to understand the origin and nature of the neutrino
> mass, as well as its absolute mass scale. I encourage you to be
> more precise with the wording here.
I have revised the old sentence "concluding the studies of the
Standard Model with three massive neutrinos" by "achieving the final
goal of studies on standard three-flavor neutrino oscillations".
> - I found muTristran an interesting idea; however, not enough
> details are given to state its experimental reach. Please add some
> benchmark numbers of muTristran performance, e.g., expected rates
> or neutrino luminosity.
Following the referee's suggestion, I have added the sentence "The
expected number of muons at $\mu$TRISTAN is on the order of $10^{13}$
to $10^{14}$ muons per second."
> - After equation (23), you say that "it's remarkable that the T
> violation is proportional to the Jarlskog factor in a vacuum."
> This is true, though the amplitude of the T-violation is still
> matter dependent since you have the ratio of the Delta E to Delta
> E\tilde. So, the extraction of the Jarlskog invariant through this
> method would still be matter-effect dependent. Please make the
> phrasing more accurate.
To make the statement clearer, I have added the sentence ", when
divided by the $\delta$-independent factor $16\,\prod_{j>k}
[\sin(\Delta \widetilde{E}_{jk}L/2)\Delta E_{jk}/\Delta
\widetilde{E}_{jk}]$".
> - I think a brief discussion (one or two sentences) on current
> constraints of CPT symmetry with neutrinos should be present in
> the introduction, since this is needed to connect T and CP
> oscillation probabilities.
Following the referee's suggestion, I added a footnote "To justify
discussions of T violation on an equal footing with CP violation, CPT
symmetry is necessary in the neutrino sector. Refs.[27-31] studied
CPT symmetry in the neutrino sector and concluded that there are
strong constraints on CPT violation.", as well as Refs.[27-31] in
References at the end.
> - I think the paper would benefit with some figures that can compare
> the results of Eqs. (23), (28), and (42) for benchmark values of
> epsilon and eta.
As I mentioned at the end of Conclusions and in the footnote on page
7, the aim of this paper is to derive the analytic form of T violation
and estimating experimental sensitivity is beyond the scope of this
paper. So quantitative discussions, such as comparisons of the
expected sensitivity with the current bounds etc., are left for future
works.
> - on the conclusion, it would be nice if the author echoed the
> experimental programs that can measure this effect.
Following the referee's suggestion, I added the words "such as
$\mu$TRISTAN or other types of neutrino factories".
I hope I have successfully addressed all the points made by the
referee and the manuscript can be accepted in its present form.
Thank you.
Osamu Yasuda
Reviewer 2 Report
Comments and Suggestions for Authors
The authors of this article give analytical formulas for T violation in neutrino oscillations in three flavor standard neutrino oscillations: a non-standard interaction case and an unitarity violation case. They note that exploring the T violation energy spectrum could be helpful in identifying novel neutrino physics phenomena in a recently proposed idea involving the acceleration of μ+ is called μTRISTAN. New designs for neutrino factories are being considered as a result of this endeavor. Furthermore, measurements of T violation in neutrino oscillations are being investigated by taking advantage of the polarization of μ+.
The analytical expression for T violation in neutrino oscillations under three distinct scenarios a flavor-dependent non-standard interaction scenario, a case with unitarity violation, and the usual three flavor mixing frameworks has been obtained in this research. Future long-baseline experiments may be able to explicitly explore unitarity in the νμ ↔ νe channel, as they observed that they attain great sensitivity to T violation across a large energy range. Additionally, they showed that, whereas one coefficient in the normal case is energy-proportional, there is an extra energy-independent contribution in the case of non-standard interactions. Thus, by studying the energy dependence of T violations, it might be possible to observe the consequences of non-standard interactions.
Regarding neutrino non-standard interaction observance in the μTRISTAN experiment, this paper is useful. It is recommended that the article be published in its existing format.
Author Response
Dear Referee,
Thank you for your positive feedback and recommendation to publish the article in its existing format. I greatly appreciate your time and effort in reviewing my manuscript.
Best regards,
Osamu Yasuda